# The Design and Validation of an Intensity-Modulated Multipoint Fiber-Optic Liquid-Level Sensor

**DOI:** 10.3390/s25165009

**Published:** 2025-08-13

**Authors:** Abdul Ghaffar, Sanku Niu, Mujahid Mehdi, Sadam Hussain, Ahmed Muddassir Khan, Zamir Ahmed Abro, Muhammad Saleh Urf Kumail Haider, Zhanyou Chang, Xiaoyu Chen, Salamat Ali

**Affiliations:** 1Key Laboratory of Air-Driven Equipment Technology of Zhejiang Province, College of Mechanical Engineering, Quzhou University, Quzhou 324000, China; 92ghaffar@gmail.com (A.G.);; 2Faculty of Design, Aror University of Art Architecture Design & Heritage Sindh, Sukkur 65200, Pakistan; 3Hangzhou International Innovation Institute, Beihang University, Hangzhou 311115, China; 4Department of Electrical Engineering, Indus University, Karachi 75300, Pakistan; 5School of Microelectronics and Communication Engineering, Chongqing University, Chongqing 400044, China; 6School of Materials and Energy, Lanzhou University, Lanzhou 730000, China

**Keywords:** liquid-level measurement, multipoint sensor, coupling method, intensity variation method

## Abstract

This study introduces a cost-effective solution and sensor arrays for the multipoint liquid-level measuring sensor based on an intensity modulation technique. The sensor structure is based on the twisting of two fibers and creates cascading to achieve a multipoint detection. Three sensors are fabricated on a single illuminated polymer optical fiber. The twisting creates side-coupling between two fibers, and the coupled power is attenuated when liquid emerges in the coupled region. Each sensor has its own output source, which is connected to the power meter. When the liquid-level increases, the coupled power is continuously decreased. The multipoint liquid-level sensor is theoretical and experimentally tested. The experimental results show that sensors have a good response and linearity. The sensors are able to measure the liquid-level up to 12 cm and have a sensitivity of about 0.2726 μW/cm, 0.1715 μW/cm, and 0.1281 μW/cm, respectively. The different flow rate (50 mL/min–300 mL/min) is also analyzed to validate the dynamic response of the sensor. The sensor demonstrates a high sensitivity and resolution in the liquid-level detection. Meanwhile, the liquid-level variation is individually and simultaneously measured. The system does not require any decoupling technique as the system relies on a single LED source, and the coupled power is individually measured from each power meter. The system represents a significant advancement in precise liquid-level sensing technology, as the system has advantages of a flexible, durable, cost-effective, and active response with respect to changes in the liquid-level.

## 1. Introduction

Fiber sensors are mainly based on silica fiber and polymer optical fiber (POF). The POF offers several advantages over traditional glass optical fibers, making it suitable for specific applications [1]. In the field of optical sensors, the POF offers several unique advantages that make it particularly suitable for certain types of sensing applications. The POF’s flexibility and larger diameter facilitate its integration into various sensor designs, allowing for more creative and compact arrangements [2,3,4,5]. This is especially useful in applications where the sensor needs to conform to complex shapes or be embedded within structures. The greater mechanical compliance of the POF compared to glass fibers enables its use in sensors that detect physical deformation, vibration, or strain [6]. By measuring changes in light transmission properties due to bending or strain, POF-based sensors can act as highly sensitive mechanical sensors. The POF has been used in different sensing applications such as displacement [7], angle [8], force [9], humidity [10], RI [11], pressure [12], liquid-level [13], and many more [14,15]. POFs can be used in multipoint sensing systems through different techniques like wavelength division multiplexing, time division multiplexing, or intensity variation, allowing for multiple sensors to be connected along a single fiber [16,17,18]. This enhances efficiency and reduces the cabling complexity.

Liquid-level sensors are crucial components in industrial, commercial, and domestic applications, where the precise measurement of fluid quantities is necessary [19]. These devices are designed to detect and monitor the level of liquids in tanks, reservoirs, or containers, ensuring the efficient management of resources, preventing overflows or emptying, and enhancing safety in various operations. However, conventional methods for measuring and monitoring liquid-levels, such as float systems [20], pressure transducers [21], hydrostatic sensors [22], ultrasonic devices [23], acoustic Doppler profilers [24], radar sensors [25], capacitance sensors [26], and float-operated mechanical setups, often lack the precision desired for an exact liquid-level determination. Conventional liquid-level sensors also have a lack of multiplexing capabilities, where a multipoint liquid-level sensor is an advanced system capable of detecting and reporting the liquid-level at several discrete points within a tank or container. Instead of providing just a high- or low-level indication, these sensors offer granular monitoring, allowing users to track the liquid-level more precisely throughout the entire range. Multipoint sensors often employ a combination of the techniques mentioned above, using multiple sensing elements or a single adaptable technology like ultrasonic or capacitive sensors with multiple measuring points [27].

Extensive research has been conducted on utilizing POFs through various techniques for liquid-level sensing applications [28,29]. A comprehensive review of a polymer optical fiber level sensor is given in reference [30]. These methodologies encompass approaches such as the side polishing of fibers and the exploitation of fiber twisting and bending configurations. Jing et al. [31] presented a multipoint liquid-level sensor structure based on the race track helical method using the POF. In effect, the sensor was a single liquid-level sensor where multiple points are created to detect the liquid. Those multiple points of liquid detect were converted to the liquid-level. Lin et al. [17] presented another method that was a discrete liquid-level system and created multiple segments which were aligned coaxially. Some other methods were also reported for multipoint liquid-level sensors, such as the lateral polishing of the bend [18], the RI modulation of the macrobend [32], and the side-coupling of LEDs [33].

In the context of multipoint liquid-level sensors, some other researchers used other terms such as multiplexed and quasi-distributed [34,35,36]. Wang et al. [35] reported a quasi-distributed liquid-level sensor based on the cross-correlation optical time domain Fresnel reflection method, where multiple gaps were created in a single mode optical fiber. In the side-coupling method [33,36], an LED-belt is employed to side-couple the light, and each LED becomes a single sensing point for the liquid detection. This method can achieve multipoint liquid detection but cannot offer multipoint liquid-level sensors. However, there is limited literature on multipoint liquid-level measurement sensors. Most reported works focus on single liquid-level sensors or multipoint liquid detection systems that ultimately achieve a single liquid-level measurement. Achieving multipoint liquid-level sensing using a single optical fiber presents a considerable challenge due to the complexity of maintaining accurate and reliable measurements across multiple sensing points. This gap in the existing literature underscores the significance and novelty of the approach proposed in this study.

The present study introduces an innovative multipoint liquid-level sensing solution utilizing the POF. This novel sensor design employs a sophisticated twisting method, wherein three secondary fibers are meticulously twisted at distinct segments along a primary fiber, thereby establishing three independent sensing points within a single illuminating fiber. Light from an LED source is injected into the primary fiber, while each secondary fiber is linked to a power meter to meticulously detect fluctuations in liquid-levels. The twisting creates an optical power coupling in the secondary fiber. In the sensor working phenomenon, the coupled power attenuates as the liquid-level increases. The structure of this study is organized systematically: Section 2 describes the sensor design, elaborating on the twisting methodology and implementation process, while Section 3 and Section 4 present the experimental procedures, results, and corresponding discussion.

## 2. Materials and Methods

### 2.1. Sensor Fabrication

The initial phase of sensor development is depicted in Figure 1. The proposed sensing system comprises three primary components: an optical fiber, a light source, and a power meter. An optical power meter (Thorlabs PM100USB with S151C photodetector, Thorlabs, Newton, NJ, USA) was used for power measurement. Thorlabs M405FP1 LED served as the light source, and polymethyl methacrylate (PMMA) fiber was used, specifically Mitsubishi’s SK-40 variant. POF is leveraged for efficient short-range transmission with minimal loss, particularly in the visible spectrum and also widely used in sensing application [37]. POFs fundamentally feature a broad core encircled by a slender cladding layer, distinguished by their respective refractive indices (RIs). Here, the cladding exhibits a lower RI compared to the core, facilitating light confinement within the fiber core. The chosen PMMA fiber boasts a core diameter of 980 µm with an RI of 1.492, while its cladding, measuring 10 µm thick, has an RI of 1.402.

Sensor fabrication involves twisting a pair of fibers, thereby enabling the transfer of radiation loss from one fiber to the other; this twisted section serves as the sensor’s active detection zone. One fiber is designated as the input, linked to an LED light source, while the other functions as the output, attached to power meters. However, cascading of twisting structure is employed to achieve a multipoint sensor system. In this study, the multipoint liquid-level sensors are termed Sensor-1, Sessor-2, and Sensor-3, respectively. Sensor-1 is represented with a green line, Sensor-2 is represented with blue line, and Sensor-3 is represented with yellow line. However, the red line refers to illuminating fiber connected with light source. All sensors have separate output ends. In this way, all sensors respond independently with liquid-level variation.

### 2.2. Sensing Principles

This section aims to provide a descriptive overview of the design structure and fundamental operational principles governing our suggested sensor. The sensor’s functionality centers around a light-coupling mechanism that employs the unique properties of POF. The schematic representation of the sensor configuration is displayed in Figure 2. It features a pair of optical fibers arranged parallel to one another with twisting, separated by a distance significantly greater than the operational wavelength (d >> λ). One fiber serves as the illumination fiber, interfacing with a light source, while the other is dedicated to detection, connecting to the photodetector. The sensor sensing phenomenon lies in a coupling strategy that capitalizes on the radiation effects originating from illuminating fiber. This radiation phenomenon facilitates an exchange of optical power between the two fibers. Specifically, light radiated from the illumination fiber is coupled by the secondary fiber through the coupling process.

It is worth noting that the efficacy of coupling is governed by factors such as the fiber’s characteristics, the separation distance “d”, and the optical attributes of the medium enveloping the fibers. Hence, liquid medium also affects the coupling. The liquid is assumed to have high light absorbance; the degree of power coupling is directly proportional to the length of the fiber segment submerged in the liquid. Consequently, the optical power received by the second fiber correlates directly with the real-time level of the liquid being monitored.

### 2.3. Theoretical Modeling

To analyze the interaction between the two multimode fibers, a model is developed to characterize the power coupling mechanism, with particular emphasis on the radiation-induced power transfer occurring in the designed configuration. Traditionally, a meticulous analysis of mode coupling between dual multimode fibers would necessitate the application of classical coupled wave theory, entailing intricate calculations [38,39]. However, considering our setup leverages the natural irregularities of the fibers to induce coupling, a more pragmatic approach involves modeling the power transfer as a consequence of a stochastic coupling function linking the two fibers. Consequently, this intricate interplay can be more succinctly characterized through coupled power equations [40], bypassing the complexity of the classical theory that demands tracking both the phase and amplitude of every mode throughout the fiber’s length.

Indeed, the utilization of coupled power equations is a well-established practice for elucidating the impacts of mode coupling within a single optical fiber or among multiple fibers, a phenomenon largely attributed to structural inconsistencies [41,42]. Scholars have extensively explored the analytical formulation of these equations, leveraging perturbation theories and statistical methodologies. It is noteworthy that existing solutions often come with specific limitations imposed by underlying assumptions. Consequently, we undertake the task of either deriving anew or adapting these solutions into a generalized form that aligns seamlessly with the requirements and objectives of our investigation. The interaction and power transfer between two parallel fibers can be rigorously modeled using a set of first-order ordinary differential equations [40,41,42]:(1)$$\left\{\begin{array}{c} \frac{d}{dx}P_{0}\left(x\right)=-\left(\alpha +\beta \right)P_{0}(x)+\beta P_{1}(x)\\ \frac{d}{dx}P_{1}\left(x\right)=\beta P_{0}\left(x\right)-(\alpha +\beta )P_{1}(x) \end{array}\right.$$
where $P_{0}\left(x\right)$ is the power in illuminating fiber; $P_{1}\left(x\right)$ is the power in secondary fiber; β is the coupling coefficient; α is power attenuation coefficient of the fibers; and *x* is the distance. Here, the attenuation is assumed to be equal because both (illuminating and secondary) fibers are taken as same type of fiber. Equation (1) expresses a straightforward, intuitive concept. To grasp its essence, Figure 2 offers an interpretative visualization through a power balance perspective, akin to a finite-difference equation paralleling (1), applied to a minuscule segment of length Δ*x*. The expression for *β*(*x*), influenced by liquid absorption, can be delineated as follows:(2)$$\beta \left(x\right)=\left\{\begin{array}{c} \beta \;\;\;\;\;\;\;\;0<x<L_{max}-L\\ 0\;\;\;L_{max}-L<x<L_{max} \end{array}\right.$$

Equation (2) is confined exclusively to the coupling zone, designated as $C=\{x:0<x<L_{max}-L$, wherein L signifies the current liquid-level, and $L_{max}$ denotes the sensor’s maximum operational range. Within the setup, the initial power $P_{in}$ is introduced into the illumination fiber at the starting point x=0, while the resultant output power $P_{out}$ is detected at the extremity of the secondary fiber located at ${x=L}_{max}$ (refer to Figure 2). By adopting the boundary conditions where $P_{0}$ (0) = $P_{in}$ and $P_{1}$ (0) = 0, the fundamental solution to Equation (1) can be compactly formulated as follows:(3)$$\left\{\begin{array}{c} P_{0}\left(x\right)=-\frac{P_{in}}{2}e^{-\alpha x}\left(1+e^{-2\beta x}\right)\\ P_{1}\left(x\right)=-\frac{P_{in}}{2}e^{-\alpha x}\left(1-e^{-2\beta x}\right) \end{array}\right.$$

Employing Equation (3), the average power within the secondary fiber, assessed at the end of the coupling distance $x=L_{max}-L$, is articulated as(4)$$P_{1}\left(L_{max}-L\right)=-\frac{P_{in}}{2}e^{-\alpha L_{max}-L}\left(1-e^{-2\beta L_{max}-L}\right)$$

Consequently, the resultant output power, or the power measured at the end of the secondary fiber *x* = *L*, can be formulated as follows:(5)$$P_{out}=P_{1}\left(L_{max}-L\right)e^{-\alpha L}$$(6)$$P_{out}=-\frac{P_{in}}{2}\left(1-e^{-2\beta (L_{max}-L)}\right)e^{-\alpha L_{max}}$$

It is important to acknowledge that within the region ${∆x=\{x:L}_{max}-L<x<L_{max}\}$, Equation (1) persists but with β=0, assuming the liquid is highly absorptive. This condition rationalizes the presence of the term $e^{-\alpha L}$ in Equation (6).

### 2.4. Parameters for Model Configuration

In this section, we adapt the previously derived analytical model formulas to incorporate typical parameters pertinent to our sensing setup. Subsequently, these theoretical formulations are subjected to numerical assessments. The attenuation coefficient α encompasses both absorption and scattering losses inherent to each fiber. Notably, in fibers absorption losses are insignificant relative to radiation losses. Given that attenuation losses in SK-40 fibers can reach substantial levels of 0.15 dB/m [43], high α values are appropriately adopted in our considerations. Regarding β, the scenario entails the illumination fiber emitting power over an extensive solid angle, with the separation d≫λ, leading to only a minor fraction of the emitted light being coupled into the detection fiber. Consequently, within our specific sensor design, we anticipate a scenario of weak coupling $\beta \ll m^{-1}$. The normalized output power, defined as $P_{out}/P_{in}(P_{out}$=$P_{out}(L))$ calculated per Equation (6), is graphed in Figure 3. It illustrates the relationship between the liquid-level L and the output power for a fixed $\beta =0.01m^{-1}$, $L_{max}=12\;cm$, and varying α values spanning from 0.5 to 3$m^{-1}$. Observably, the output power demonstrates a linear correlation with the liquid-level, irrespective of the fiber’s attenuation. This observation stems from the circumstance that when β≪1 is sufficiently small, Equation (6) can be approximated to yield a linear relationship, reflecting this simplified dynamic.

## 3. Experimental Setup

The sensors were integrated into a container at different positions, as shown in Figure 4. The measurement was taken from power meters 1, 2, and 3. The input optical power was about 20 mW, which was launched from the LED source. All sensors were embedded within a water container. Different configurations could be analyzed, such as sensors positioned at the same level or sensors positioned at different levels, as well as simultaneous and individual responses. Initially, all sensors were positioned at the same elevation, resulting in an equal variation in the liquid-level, as depicted in Figure 4. Conversely, in the second scenario, an inclination was created to analyze different liquid-level responses at the same time. A pump motor was used to inject water, and different flow rates were also injected.

### Spectral Response of Sensor

The experimental setup was designed to evaluate the spectral performance of the liquid-level sensors. It involves a light source which contains an LED operating at 635 nm. This specific wavelength is within the visible spectrum, offering efficient light coupling and propagation in polymer optical fibers. An Ocean Optics 2000+ (Ocean Insight, Orlando, FL, USA) spectrometer is used for precise spectral measurements. The spectrometer measures the intensity of the light that exits the fibers (referred to as the “photon count” in Figure 5) over a range of wavelengths. The multipoint liquid-level sensor system uses twisted fibers, where light coupling occurs between two fibers. The fiber through which light moves from the source is designated as the illuminated fiber. The fibers where the light coupled from the illuminated fiber is measured are designated as receiving fibers. Light from the LED is launched into the illuminated fiber. The coupled light intensity is collected from the receiving fibers and analyzed using the spectrometer. Measurements are conducted under two distinct conditions for each sensor: (1) without water (dry condition), where the sensor is exposed to air, and (2) the in-water condition, where the sensor is fully immersed in water.

When the sensors are immersed in water, the photon count (intensity) decreases significantly compared to the dry condition. This is evident from Figure 5, where the intensity curves for “in-water” conditions (e.g., red, green, and orange curves) are lower than their corresponding “dry” conditions (e.g., black, blue, and purple curves). The spectral characteristics do not exhibit a significant wavelength shift, but the intensity is strongly influenced by the external medium. The reduction in the photon count when the sensors are immersed in water is attributed to the change in the RI medium. Water has a refractive index closer to that of the polymer fiber, which reduces the total internal reflection within the fiber. Consequently, more light escapes from the fiber, resulting in a lower coupled light intensity in the receiving fiber. The immersion in water increases the light loss due to absorption and scattering at the interface. The twisted geometry of the fiber likely enhances the sensitivity to external changes in the surrounding medium (air vs. water). This is because twisting can exacerbate the light leakage when the refractive index of the surrounding medium changes. The results demonstrate that the spectral response of the sensor is sensitive to changes in the surrounding medium, making it suitable for the liquid-level detection.

## 4. Results and Discussion

When designing a sensor that incorporates a liquid, it might be crucial to examine properties like the refractive index, the viscosity, and the liquid’s compatibility with polymer optical fibers. The different types of liquids are compatible with the POF, such as glycerol, silicone oil, mineral oil, ethanol, and water. The water is easily accessible and possesses a lower refractive index, making it suitable for use in experiments where the sensor does not require sealing against moisture. In the context of our research, water was the chosen medium, although other fluids could be suitable alternatives. The multipoint liquid-level sensor was tested for monitoring and measuring liquid-levels at various points dynamically. The system enables exact and simultaneous recordings of the liquid-level across different points. The data is illustrated graphically, showing the relationship between the coupled intensity (measured in Watts) and the liquid-level (expressed in centimeters, cm).

Figure 6 illustrates the sensor’s performance during sequences of the ascending and descending of liquid-levels. In this setup, three sensor probes, labeled Sensor-1, Sensor-2, and Sensor-3, are submerged in water to measure the liquid-level. Variations in the liquid-level are inferred from changes in the optical power intensity. Specifically, Figure 6a exhibits the sensor’s reaction to steadily ascending levels following the injection. Here, the vertical axis signifies the peak optical power recorded at the reference point and considered as a zero liquid-level. As the liquid-level ascends, the sensors show alterations through variations in the optical power intensity; the immersion of the liquid causes an attenuation in the coupled intensity. These interactions alter the attenuation around the probes, thereby progressively diminishing the power intensity as the liquid-level rises.

It is important to note that the initial and final levels of coupled power can be influenced if the specific twisting conditions of the sensors changed, but once the sensor is fabricated, then the coupled power will be the same at the initial and final liquid-level. Nonetheless, employing the same probe repeatedly yields consistent initial and final power levels. The initial coupled power was 3.65 μW, 2.23 μW, and 1.66 μW; meanwhile, the final coupled power (at 12 cm level) was 0.38 μW, 0.24 μW, and 0.12 μW for Sensor-1, Sensor-2, and Sensor-3, respectively. The initial couple power slowly decreased from Sensor-1 to Sensor-3. This is due to the illuminating power, which is higher at the first twisting, and then it slowly decreased along the length of the fiber. However, the fundamental response pattern of all sensors remains consistent: an optical power decrease with an increase in the liquid-level. Figure 6b illustrates the sensor’s behavior when the liquid-level is decreasing, as water is ejected from the container. This reduction in the liquid-level induces a change in the attenuation, resulting in an increase in the coupled optical power since less of the sensor’s probe is submerged in water. A visual inspection reveals that the sensor correspondingly amplifies its power intensity in direct proportion to the decline in the liquid-level.

During the descending of the liquid-level, the sensors exhibit a gradual rise in the optical power intensity. Notably, under conditions of receding liquid-levels, the sensors demonstrate a tardier response time compared to instances when the liquid-level is ascending. This disparity primarily stems from the sensors’ wettability characteristics. The sensors become saturated during the influx of liquid, leading to a sluggish alteration in the reflective index as the liquid recedes. To mitigate this discrepancy and ensure a uniform sensor responsiveness irrespective of whether the liquid-level is rising or falling, a potential solution involves chemically treating the sensors with a hydrophobic agent. This modification would counteract the delayed response dynamics observed during the liquid-level depletion.

The performance of the sensors in measuring liquid-levels is illustrated in Figure 7, utilizing data collected from three separate power meters. As depicted, all sensors exhibit a linear response pattern as the liquid-level ascends, with each capable of detecting liquid up to a 12 cm depth, a limitation set by the fabrication length of the sensor probes. Accompanying the data in Figure 7 are fitted curves, seemingly representing linear regression models, which closely fit the respective data sets for each sensor. The graph also discloses the equations of these lines alongside their associated R-squared coefficients, evidencing the fitness of the linear model to the collected data. Specifically, Sensor-1 adheres to the equation y = −0.3x + 3.9 with an R-squared of 0.9979; Sensor-2 follows y = −0.19x + 2.5 with an R-squared of 0.9946; and Sensor-3 operates according to y = −0.14x + 1.8 with an R-squared of 0.9933. These near-unity R-squared values affirm a robust linear correlation between the liquid-level and measured intensity across the sensors. The measured results have similar responses to the theoretical results. The calculated sensitivities for these sensors stand at 0.2726 μW/cm, 0.1715 μW/cm, and 0.1281 μW/cm, respectively, revealing a trend of a declining sensitivity with an increase in the number of sensors attached to the illuminating fiber. Extrapolating from this pattern, it is inferred that introducing additional sensors beyond the current setup (Sensor-4 onwards) would likely yield an even lower sensitivity compared to Sensor-1. The sensors’ overall performance parameter is shown in Table 1. Nevertheless, the observed inverse relationship between the liquid-level reduction and the optical power intensity increment furnishes a reliable methodology for the surveillance and quantification of liquid-levels across diverse scenarios, spanning from storage tanks to transportation systems.

Figure 8 presents a series of repetitive sensor responses to the continuous rise in the liquid-level. The test for the liquid-level sensing repeatability was conducted at three distinct time intervals. Observations from the experiment revealed a striking consistency and similarity in the sensor’s responses, regardless of whether the liquid-level was ascending or descending. This recurrent sensor response curve vividly underscores the sensor’s dependable sensing function across varied timeframes. The small variation in the measurement response is negligible. This small variation may be caused by water motion that might increase or decrease the level. In aggregate, the repetitive sensor response curves depicted underline the system’s proficiency in consistently delivering repetitive and dependable outputs in response to fluctuations in liquid-levels, thereby reinforcing its reliability and precision in liquid monitoring applications.

Figure 9 illustrates the sensor’s response under varying flow rates, validating its capability for dynamic responses. For the dynamic response, a pump motor was used to generate a continuous water flow at varying rates, simulating real-time liquid-level changes. This configuration enabled the evaluation of both the response speed and tracking capability. The measured response time was approximately 0.1 s, determined by the sampling rate of the power meter, indicating the sensor’s suitability for rapid detection. In this flow rate examination, four distinct rates (50 mL/min, 100 mL/min, 200 mL/min, and 300 mL/min) were assessed. Elevated flow rates expedite the rise in the liquid-level, and, consequently, the sensors promptly detect these alterations. Notably, the sensor adeptly senses swift changes in the liquid-level induced by rapid flow rates. At the uppermost flow rate of 300 mL/min, a conspicuous and rapid decrease in the coupled power intensity is observed, indicative of the sensors’ prompt adjustment to heightened liquid-levels. Conversely, lower flow rates, like 50 mL/min, cause a gradual submerging of the sensors and a slow rise in the liquid-level, which the sensor effectively measures, highlighting a steady increase in the liquid-level attributed to a slow flow. This analysis compares the sensor’s dynamic response across a range of flow rates, with the sharp decline in the power intensity at peak flow rates emphasizing the sensor’s proficiency in rapidly adapting to altering liquid-levels. Simultaneously, the more leisurely decrease at minimal flow rates attests to the sensor’s adaptability to accommodate any flow rate as per specific demands. Collectively, the sensor’s reactions across disparate flow rates suggest that the proposed multipoint liquid-level sensor is versatile, functioning effectively across a range of velocities while consistently producing response patterns, thereby affirming its suitability for diverse operational contexts.

Thus far, our examinations have been confined to scenarios where the liquid-level uniformly increases or decreases in the container, ensuring that every sensor encounters equal liquid-level alterations simultaneously. Our goal was to develop a system with multiple sensors integrated into a single illuminating fiber which should be capable of detecting random variations in the liquid-level simultaneously across different sensors. To accomplish this objective, it is essential to evaluate the sensors’ response across a range of distinct liquid-level scenarios. Two experimental approaches were considered to effectively execute this task. In the first, three separate containers are carried out, and each sensor is placed in its own separate container. The liquid-level in each container is raised or lowered independently. This approach required three separate containers, with one sensor in each container. But this method increases the complexity of the experimental setup, as it requires more containers and a mechanism to control the liquid-level in each container individually.

The second approach involves placing all sensors within a single container but inclining the container. This orientation naturally creates a slope of liquid-levels across the container, with one end having a higher level than the other. By doing so, the sensors are simultaneously exposed to varying liquid-levels, enabling the study of their collective response under diverse conditions, like real-world scenarios. We adopted this method for our experiment. By inclining the container, we created a slope of liquid-levels, allowing us to analyze the response of each sensor to different liquid-level conditions simultaneously. In our experiment, the container was filled to half its capacity. We designate a 6 cm liquid-level as our reference point. By inclining the container, one end will have a liquid-level higher than 6 cm, while the opposite end will have a lower level. This setup enables us to observe how each sensor responds to different liquid-levels within a single container.

Figure 10 illustrates the experimental setup. It shows the inclined container with sensors placed at different positions. Each sensor is exposed to a different liquid-level, allowing us to simultaneously analyze their responses to varying conditions. The inclination angle was approximately 45°. This inclined container approach not only simplifies the experimental setup compared to using multiple individual containers but also introduces a more realistic and challenging testing scenario that better mimics environments where sensors might need to operate simultaneously under differing liquid heights. It allows for the efficient evaluation of the sensors’ ability to discern and respond accurately to non-uniform liquid-level changes, a critical feature for advanced monitoring systems in industries like chemical processing, water management, or industrial automation.

Figure 11 illustrates the sensors’ response under non-uniform liquid-level conditions. Upon inclining the container, the liquid assumes a sloping configuration, creating a slope of distinct liquid-levels across the positions of the sensors. Consequently, one end of the container encounters a higher liquid-level, while the other end experiences a lower level. The liquid-level around each sensor changes differently in this inclined setup. Sensor-1 was placed at the lower end of the inclined container, where the liquid-level was higher. As the liquid-level increases, the optical intensity detected by Sensor-1 decreases. This happens because a higher liquid-level typically attenuates the light more, leading to a reduction in the optical intensity. Therefore, the optical intensity for Sensor-1 reached its minimum point under these conditions. Sensor-2 and Sensor-3 were placed towards the higher end of the inclined container, where the liquid-level was lower. The intensity was increased when the liquid-level surrounding these sensors was reduced. This phenomenon arises due to the reduced light attenuation caused by the lower liquid-level, leading to a greater optical intensity being measured. Specifically, Sensors-2 and -3 both exhibited heightened optical intensity levels as a direct result of this decrease in the surrounding liquid-level.

In the case of Sensor-1, the ascending liquid-level induced by the container’s inclination causes an intensified light attenuation effect, which subsequently reduces the measured optical intensity. This observation is consistent with the established principle that heightened liquid-levels generally exacerbate optical loss. For Sensor-2 and Sensor-3, the decreasing liquid-level leads to reduced light attenuation, consequently the measured optical intensity was increased. This increase occurs because the reduced liquid surrounding the sensors minimizes interference with light propagation, thereby boosting the signal strength. This configuration effectively illustrates the individual response of each sensor to the varying liquid-level within the same container. Inclining the container creates a distinct level for every sensor, enabling us to simultaneously evaluate their response patterns.

This system has advantages of flexible, durable, cost-effective, and active responses with respect to changes in the liquid-level. For our design, the estimated total cost of the sensing unit is USD 4.60–13.00, highlighting the affordability of the proposed approach. The sensors’ response was to measure the liquid-level variation by the different amount of optical power coupling due to the change in the reflect index. The main advantages of the present designed multipoint liquid-level sensors are to measure and monitor the liquid-level at various positions inside the system designed. Moreover, the use of the POF in the sensor design also has a number of advantages, such as being inexpensive, lightweight, and resistant to electromagnetic interference. Regarding the temperature effect, the POF-based sensors are suitable within a temperature range of 20–55 °C [44]. The current sensor design has several limitations, including a restricted sensing length, a dependency on the input optical power, and the limited chemical compatibility of the POF with aggressive fluids. Potential future developments could focus on extending the sensing range, enhancing multiplexing capabilities, integrating the system with low-cost photodetectors or smartphone-based readouts, and adapting the sensor for use with different fluid types or in harsh environments. Overall, the multipoint liquid-level sensor system developed in this work represents a significant advancement in liquid-level sensing technology.

## 5. Conclusions

This research has presented a novel approach for multipoint liquid-level sensing on a single illuminated fiber. The multipoint liquid-level sensor is based on intensity modulation techniques and deploying sensor arrays. This study focused on a sensor design involving the strategic twisting of fibers and inducing cascading to develop multipoint detection capabilities. By fabricating three sensors on a single illuminated POF, this approach achieved individual and indigenously liquid-level measurements without necessitating complex decoupling mechanisms. This system relied on one LED source and separate power meter readings for each sensor, thereby simplifying the system architecture. Theoretical and experimental validations confirmed the efficacy of the proposed sensor system, demonstrating a high responsiveness, linearity, and resolution. The sensors were capable of precisely gauging liquid-levels up to 12 cm, exhibiting sensitivities of 0.2726 μW/cm, 0.1715 μW/cm, and 0.1281 μW/cm, respectively. Beyond these quantitative metrics, the system’s adaptability, resilience, affordability, and prompt sensitivity to liquid-level changes constitute key advancements in the field of multipoint liquid-level sensing technology. Ultimately, the multipoint liquid-level sensor system developed herein offers a significant leap forward, combining practicality with precision. Its attributes of flexibility, durability, cost-effectiveness, and dynamic responses make it a compelling option for a wide array of applications where accurate and simultaneous liquid-level monitoring is paramount. In future, the tapper method will be introduced to analyze the sensitivity response of the sensor.

## Figures and Tables

**Figure 1 sensors-25-05009-f001:**
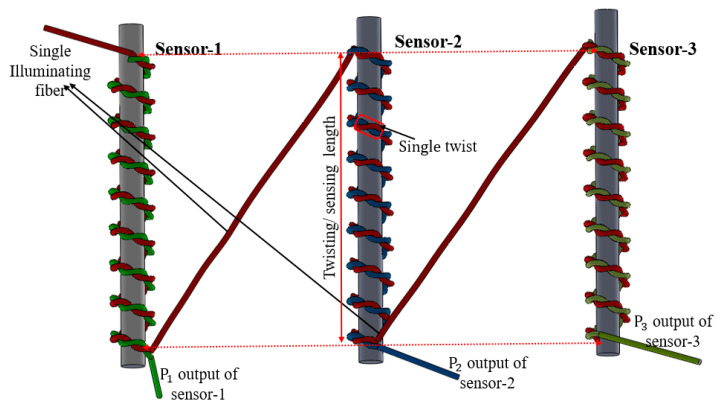
The schematic diagram of the multipoint liquid-level measurement sensor.

**Figure 2 sensors-25-05009-f002:**
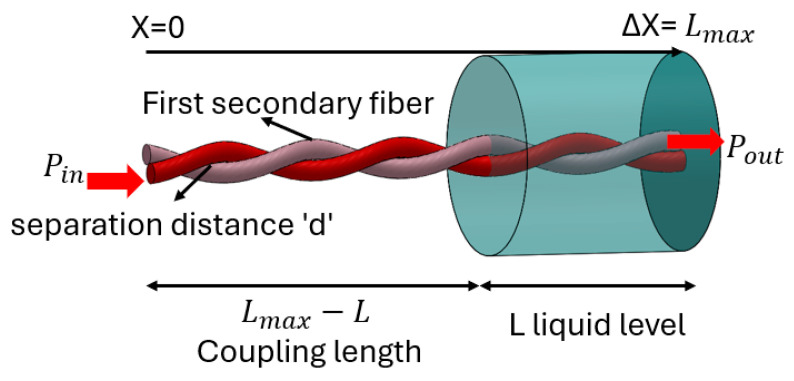
Schematic diagram of single liquid-level sensor configuration.

**Figure 3 sensors-25-05009-f003:**
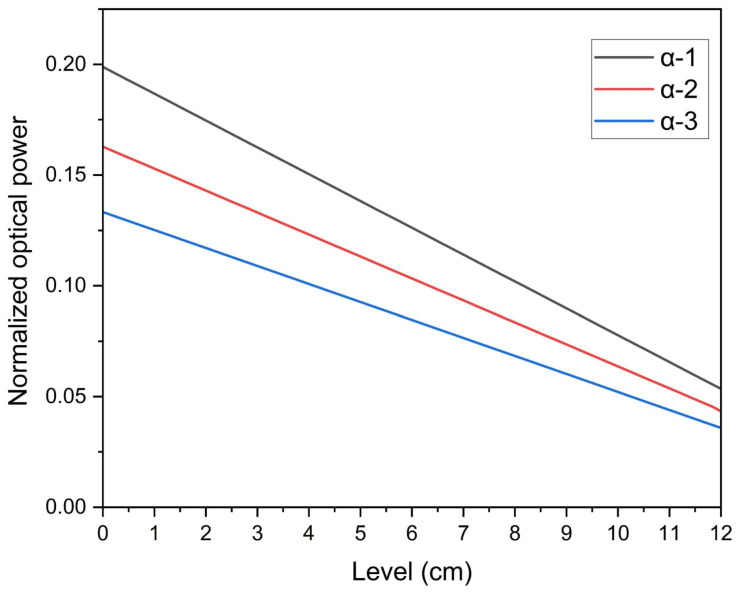
Theoretical results of liquid-level vs. normalized optical power due to coupling effect.

**Figure 4 sensors-25-05009-f004:**
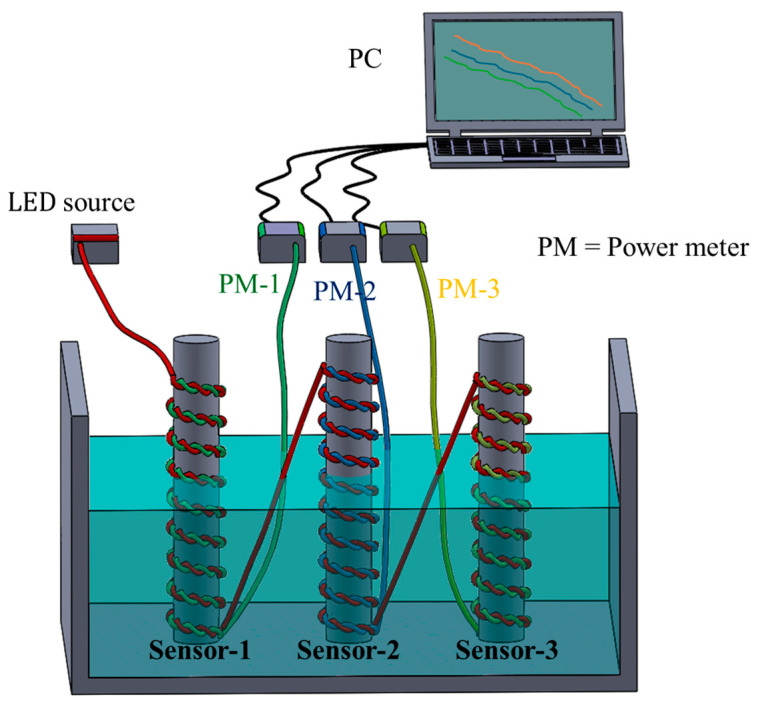
Experimental setup with multipoint sensor integration into setup.

**Figure 5 sensors-25-05009-f005:**
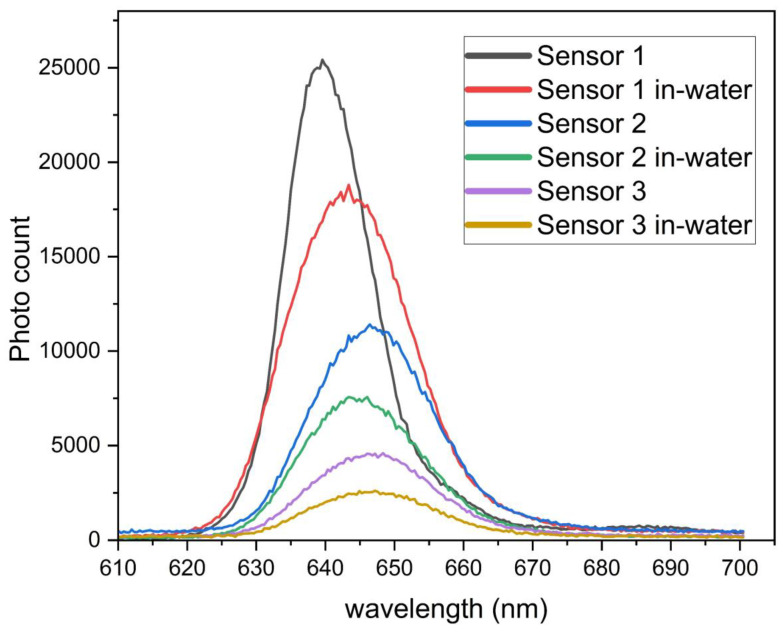
The spectral response of the multipoint sensor system in the surrounding medium (air vs. water).

**Figure 6 sensors-25-05009-f006:**
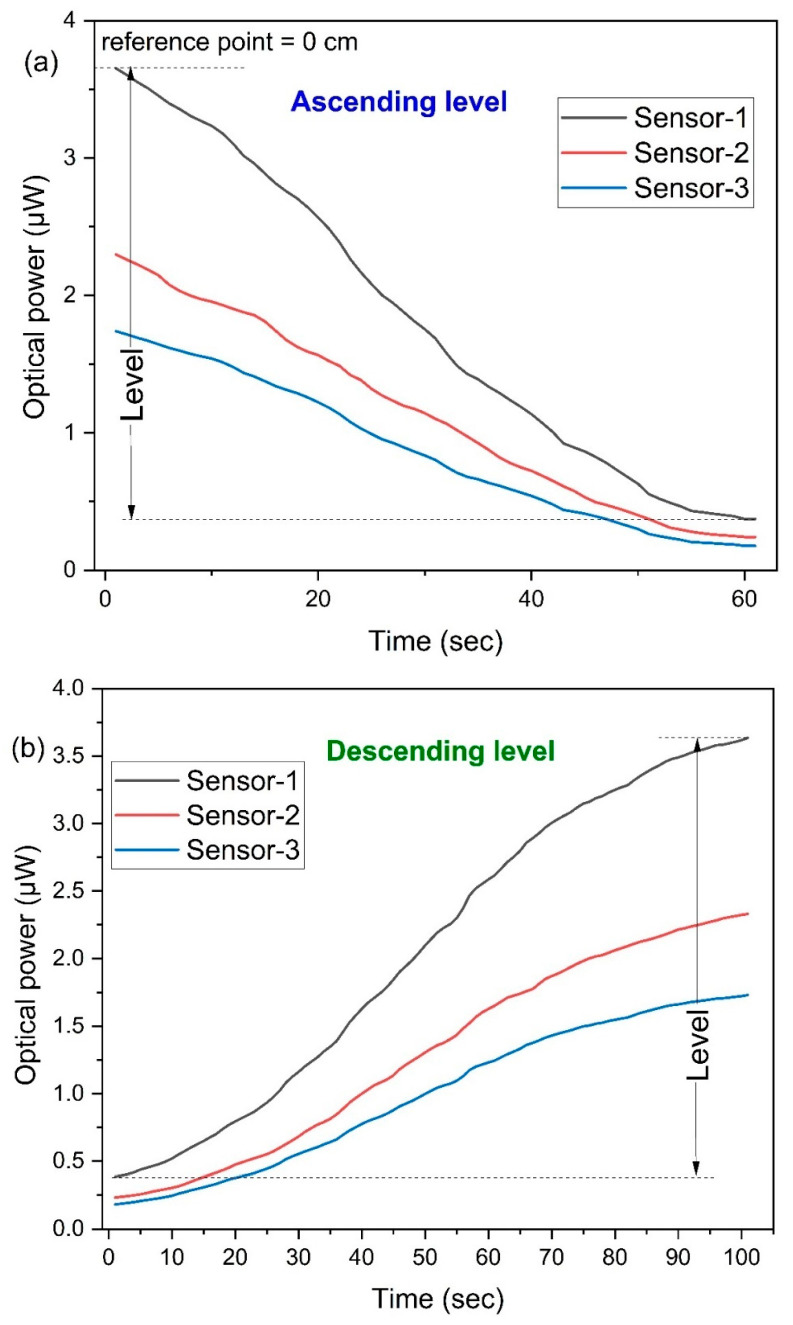
Multipoint liquid-level sensor response with liquid-level vs. time measurement: (**a**) ascending liquid-level and (**b**) descending liquid-level.

**Figure 7 sensors-25-05009-f007:**
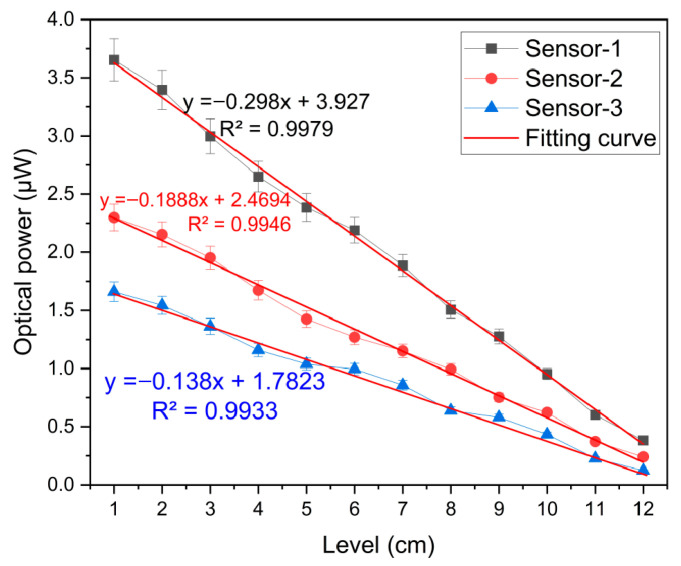
The measurement response of the multipoint liquid-level sensor.

**Figure 8 sensors-25-05009-f008:**
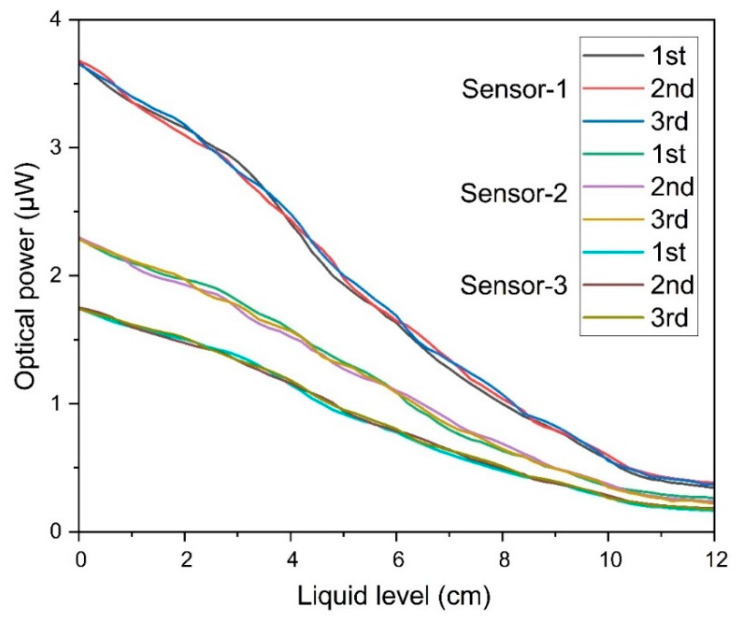
The repeatability response of the sensor for multiple measurements.

**Figure 9 sensors-25-05009-f009:**
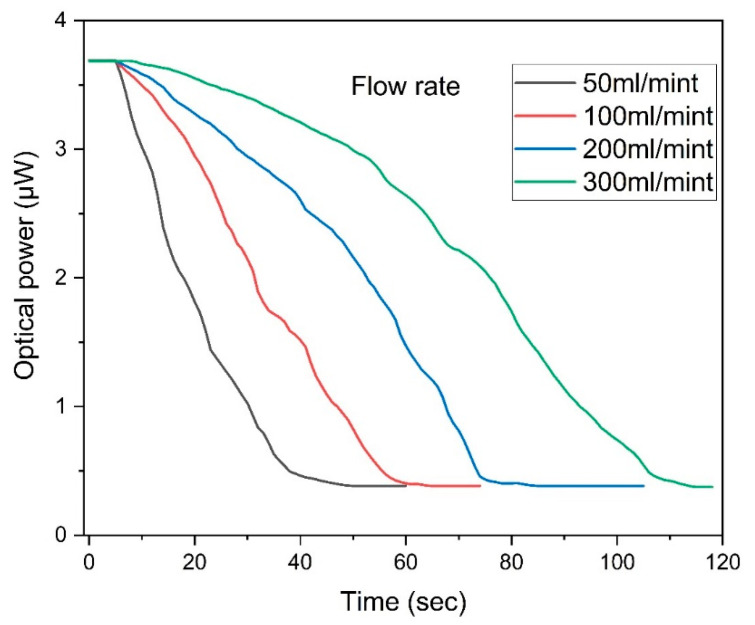
Analyzing the flow rate’s effect on the sensor’s response.

**Figure 10 sensors-25-05009-f010:**
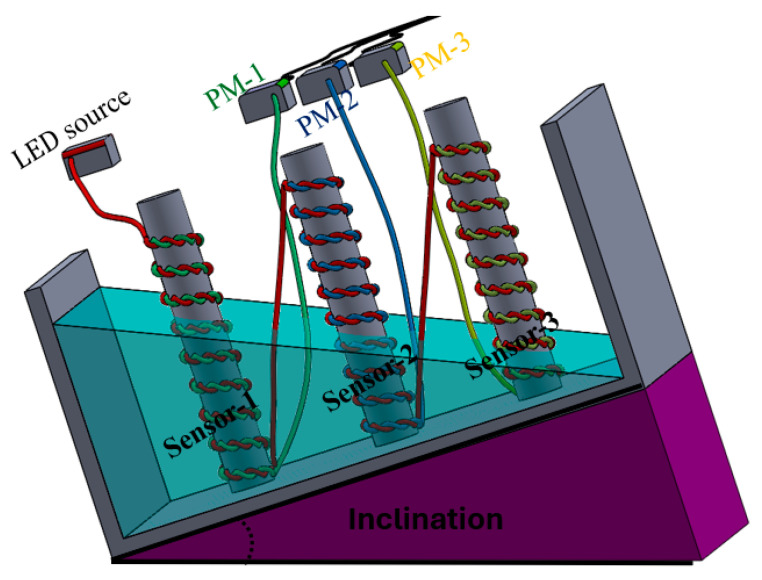
Illustration for different liquid-levels during inclination or dynamic motion.

**Figure 11 sensors-25-05009-f011:**
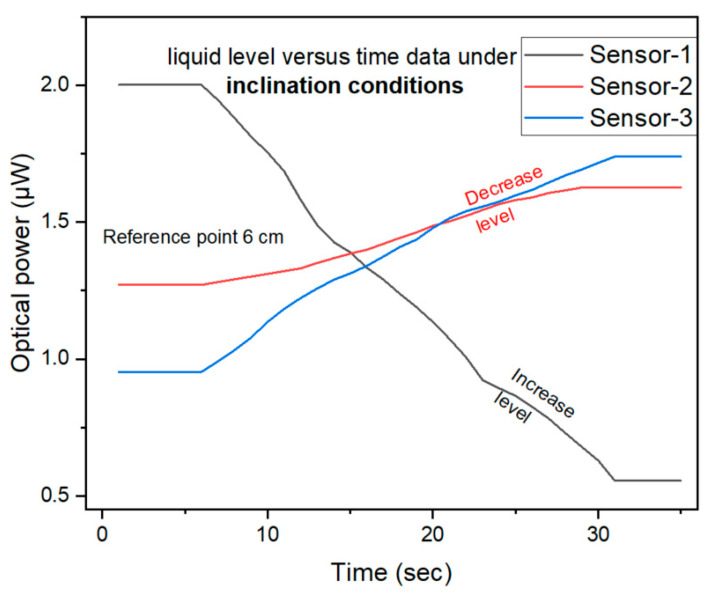
Results of different liquid-levels during inclination situations.

**Table 1 sensors-25-05009-t001:** Overall performance parameter of multipoint liquid-level sensor.

No.	y=	Sensitivity	R^2^	Resolution
Sensor-1	−0.298x + 3.927	0.2726 μW/cm	0.9979	3.7 μm
Sensor-2	−0.188x + 2.4694	0.1715 μW/cm	0.9946	5.8 μm
Sensor-3	−0.138x + 1.7823	0.1281 μW/cm	0.9933	7.8 μm

## Data Availability

The data that support the findings of this study are available from the corresponding author upon reasonable request.

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
