# Peer review of "The Design and Validation of an Intensity-Modulated Multipoint Fiber-Optic Liquid-Level Sensor"

_sensors, 2025, doi:10.3390/s25165009_

Round 1
Reviewer 1 Report
Comments and Suggestions for Authors
This work describes s a cost-effective solution and sensor arrays for the multipoint liquid-level measuring sensor based on intensity modulation technique.The sensor structure is based on twisting of two fiber and create cascading to achieve a multipoint detection The sensors are able to measure liquid-level up to 12 cm and have sensitivity about 0.2726μW/cm, 0.1715μW/cm, and 0.1281μW/cm, respectively. The different flow rate (50 ml/min–300 ml/min) is analyzed to validate dynamic response of sensor. Below are some questions and the detail revision points for the authors' consideration.
- In 1. Sensor fabrication, the main parametersof three main components: optical fiber, light source, and power meters need to be provided.
- In Figure 1, the main parametersof Multipoint liquid-level measurement sensor need to be provided, such as winding intervals, total length, types and elastic modulus of the two fibers, etc.
- In 2.2 Sensing Principles, there should be a formula involved.
- In 3 Theoretical modellingand 2.4 Parameters for Model Configuration, the simulation results should be presented. And it also points out what kind of assistance it will provide for the subsequent experiments.
- In 3. Experimental setup, this part of the text is incorrect and needs to be revised. And it should not have any textual repetition with the content in Section 2.1 above.
- In Figure 7 to Figure 11, the reliability and repeatability of the data have not been described. No error bar.
- The conclusion and discussion sections are missing. Please add them.
This work describes s a cost-effective solution and sensor arrays for the multipoint liquid-level measuring sensor based on intensity modulation technique.The sensor structure is based on twisting of two fiber and create cascading to achieve a multipoint detection The sensors are able to measure liquid-level up to 12 cm and have sensitivity about 0.2726μW/cm, 0.1715μW/cm, and 0.1281μW/cm, respectively. The different flow rate (50 ml/min–300 ml/min) is analyzed to validate dynamic response of sensor. Below are some questions and the detail revision points for the authors' consideration.
- In 1. Sensor fabrication, the main parametersof three main components: optical fiber, light source, and power meters need to be provided.
- In Figure 1, the main parametersof Multipoint liquid-level measurement sensor need to be provided, such as winding intervals, total length, types and elastic modulus of the two fibers, etc.
- In 2.2 Sensing Principles, there should be a formula involved.
- In 3 Theoretical modellingand 2.4 Parameters for Model Configuration, the simulation results should be presented. And it also points out what kind of assistance it will provide for the subsequent experiments.
- In 3. Experimental setup, this part of the text is incorrect and needs to be revised. And it should not have any textual repetition with the content in Section 2.1 above.
- In Figure 7 to Figure 11, the reliability and repeatability of the data have not been described. No error bar.
- The conclusion and discussion sections are missing. Please add them.
Reviewer 2 Report
Comments and Suggestions for Authors
The authors provide a very intersting work regarding an intensity-modulated liquid sensor based on optical fiber technology with an easy interrogation section (just a LED and a power meter). The behavior seems to be linear with the optical power, taking advantage of the crosstalk between the fiber, twisting each other and obtaining the sensor here investigated.
I suggest some comments to further improve the study and make it suitable for pubblication.
Figure 2 is improvable with a clearer schematic, indicating the separation distance d. Also with higher quality.
All the numerical plot lacks of quality, I suggest to improve.
The authors say that the peak shift (as in figure 5) in presence of water is very slight. But from a theoretical point of view why does it happen?
It would be intersting to calculate the uncertainty of the sensor response, also comparing the linear with high order fit (as cubic).
Authors provide repeatibility response but also reproducibility would be intersting to investigate.
Which is the behavior and how does the temperature affect the sensor response?
Which is the behavior with more dense or coloured liquids?
Could the author explain the difference of the proposed manuscript with the following one?
Rajamani, Allwyn S., M. Divagar, and V. V. R. Sai. "Plastic fiber optic sensor for continuous liquid level monitoring." Sensors and Actuators A: Physical 296 (2019): 192-199.
Reviewer 3 Report
Comments and Suggestions for Authors
This manuscript proposes a cost-effective sensor-array solution for multipoint liquid-level measurement using an intensity-modulation technique, which offers potentially valuable insights. However, several concerns remain regarding methodological rigor, clarity, and scientific robustness. Therefore, major revisions are recommended before the manuscript can be further considered for publication.
Major revisions:
-Please revise the abstract to include technical details of the experimental setup, such as calibration methods, evaluation of measurement accuracy, and assessment of system precision. Also, please specify the sensors’ resolution.
-In the introduction, please broaden the applications of POF-based sensors by including temperature sensing, with supporting references. When stating on line 51 that “POFs can be used in multipoint sensing systems through different techniques like wavelength division multiplexing or time division multiplexing or intensity variation, allowing for multiple sensors to be connected along a single fiber,” please add corresponding citations.
-Please reformulate line 88—“However, to the best of our knowledge, we didn’t find much literature about multipoint liquid-level measurement sensors”—in a more rigorous and quantitative manner.
-In Figure 2, please add the separation distance ‘d’, the segment length ‘Δx’, and indicate the illuminating fiber.
-In Section 2.1, please include technical details on the optical and physico‑chemical properties of the POF used (e.g., multimode vs. single‑mode), and specify the technical specifications of the power meter.
-The sentence on line 139—“The sensor sensing phenomenon lies in a coupling strategy that capitalizes on the radiation effects originating from illuminating fiber.”—requires more detailed explanation of the coupling strategy.
-The materials and methods section (line 237) includes: “Conversely, in the second scenario, an inclination was created to analyze different liquid level response at the same time. A pump motor was used to inject the water and different flow rate was also injected.”. Please specify the inclination angle and the flow rates used.
-Please analyze and discuss more deeply the sensor’s metrological properties, both static and dynamic (for example, time response).
-Since wettability influences measurements, please investigate and report the wettability properties of the sensor materials in more detail.
-Because the abstract refers to a “cost-effective” solution, please specify the actual costs of fabricating the sensors.
-What are the main limitations of the presented and fabricated sensor? For instance, can it operate with different types of fluids, or is it limited to highly absorptive liquids? What is the chemical resistance of the sensor materials against various fluids?
-What are the physical dimensions of the fabricated sensors? Please specify.
-Each sensor is described as capable of detecting liquid up to a 12 cm depth, a limitation imposed by the fabrication length of the sensor probes. Please comment on how this limitation affects potential applications.
-In the tilted-container configuration, in addition to presenting optical power vs. time, please include the corresponding liquid-level vs. time measurement.
-Results shown in Figures 6, 7, 8, 9, and 11 appear to be single measurements. Please provide averages from multiple trials along with dispersion values (e.g., standard deviation or measurement uncertainty). In Figure 7, alongside the fitted curves, please include a residuals plot.
-The discussion is currently brief and merged with the results; please expand it by addressing limitations, situating the work within current literature, and outlining potential future developments.
-In the current manuscript, there is no concluding section. Please add a dedicated conclusion summarizing the main findings, advantages, limitations, and suggestions for future work.
-The English is unclear in several places and needs improvement. For example:
- “The sensor also exhibits a high-resolution sensor.”
- “Various researchers have extensively explored various techniques to exploit POF for liquid-level sensing applications.”
- “The study unfolds systematically, with the second section delving into the details of the sensor's design explicates the twisting methodology, and implementation procedures.”
Additionally, correct typos like “Sensor-1, Sessor-2 and Sensor-3 respectively” (line 127) and “Sensor-B” (line 286). The phrase on line 154—“Establishing a model to depict the power coupling between the two multimode fibers—specifically, the power exchange triggered by the radiation loss in our design.”—is unclear; please rephrase. Also, clarify whether the sentence “Infect, sensor was single liquid-level sensor where multiple points are created to detect liquid” is correct.
Minor revisions:
-I suggest using the symbol “µm” instead of writing “micrometers.”
-Please use the correct unit symbol for minutes: “min” instead of “mint”.
-In Figure 1, the yellow line marking Sensor‑3 is difficult to see. Please correct this.
Comments on the Quality of English Language
The English is unclear in several places and needs improvement. For example:
- “The sensor also exhibits a high-resolution sensor.”
- “Various researchers have extensively explored various techniques to exploit POF for liquid-level sensing applications.”
- “The study unfolds systematically, with the second section delving into the details of the sensor's design explicates the twisting methodology, and implementation procedures.”
Additionally, correct typos like “Sensor-1, Sessor-2 and Sensor-3 respectively” (line 127) and “Sensor-B” (line 286). The phrase on line 154—“Establishing a model to depict the power coupling between the two multimode fibers—specifically, the power exchange triggered by the radiation loss in our design.”—is unclear; please rephrase. Also, clarify whether the sentence “Infect, sensor was single liquid-level sensor where multiple points are created to detect liquid” is correct.
Round 2
Reviewer 1 Report
Comments and Suggestions for Authors
This version can be accepted.
Author Response
We are deeply grateful for your thoughtful and supportive feedback.
Reviewer 3 Report
Comments and Suggestions for Authors
The manuscript has been improved only partially and addresses the present reviewer’s concerns only in part. Therefore, both major and minor revisions are still recommended to strengthen the technical content and improve presentation.
Major revisions:
- Please explicitly report the resolution value achieved in liquid‑level detection by the sensors, as this is critical for evaluating performance.
- The authors mention that a deeper analysis of the sensors’ metrological properties, including static and dynamic characteristics, has been added. However, it remains unclear where this new section appears in the manuscript, and where the measured response time (approximately 0.1 seconds) is included. Please clarify which section contains this analysis and ensure the response time is clearly identified.
- Additional detail is needed on how the wettability properties of the sensor materials were investigated and characterized. Please describe the experimental procedures, sample preparation, measurement techniques, and analysis methodology used.
- Please include in the manuscript the details provided regarding the justification for cost-effectiveness.
- The authors indicate they have added liquid‑level versus time data corresponding to the optical power versus time measurements in the tilted‑container configuration. However, Figure 11 does not seem to display this change. Please update Figure 11 to include the liquid‑level versus time data under inclination conditions, matching this description in the text.
- Regarding Figure 7, the authors state that both a residuals plot and measurement uncertainty have been added to improve reproducibility and quality. The method for calculating uncertainty is not described, nor is the number of repeated experiments specified. Please clarify the uncertainty estimation approach and the number of replicates performed. The residuals plot should also be provided as a separate figure, showing residuals (observed minus predicted values) on the vertical axis and either predicted values or the independent variable on the horizontal axis.
Minor revisions:
- Please correct the English in the sentence on line 108, which currently reads: “The sensor optical three main components: optical fiber, light source, and power meters.” This sentence should be revised for clarity and correct grammar.
- Although Figure 2 was updated in the Response to Reviewers document, the updated version does not seem to appear in the revised manuscript file. Please ensure that the corrected version of Figure 2 is included in the manuscript.
- The text in lines 450–455 reads like bullet points or list entries without a main verb, resulting in incomplete sentences. Please restructure those lines into complete, grammatically correct sentences for better readability.
Comments on the Quality of English Language
The English should be improved to more clearly express the research, for instance:
-Please correct the English in the sentence on line 108, which currently reads: “The sensor optical three main components: optical fiber, light source, and power meters.” This sentence should be revised for clarity and correct grammar.
-The text in lines 450–455 reads like bullet points or list entries without a main verb, resulting in incomplete sentences. Please restructure those lines into complete, grammatically correct sentences for better readability.
